# Yeast Diversity in Honey and Pollen Samples from Stingless Bees in the State of Bahia, Brazil: Use of the MALDI-TOF MS/Genbank Proteomic Technique

**DOI:** 10.3390/microorganisms12040678

**Published:** 2024-03-28

**Authors:** Raquel Nunes Almeida da Silva, Karina Teixeira Magalhães-Guedes, Rogério Marcos de Oliveira Alves, Angélica Cristina Souza, Rosane Freitas Schwan, Marcelo Andrés Umsza-Guez

**Affiliations:** 1Food Science Postgraduate Program, Faculty of Pharmacy, Federal University of Bahia, Salvador 40170-100, Brazil; raquel.nasil@gmail.com; 2Federal Institute of Education, Science and Technology of Bahia—IFBaiano, Catu 48110-000, Brazil; eiratama@gmail.com; 3Department of Biology, Microbiology Sector, Federal University of Lavras, Lavras 37200-900, Brazil; angelicacsouza.acs@gmail.com (A.C.S.); rschwan@ufla.br (R.F.S.); 4Biotechnology Department, Federal University of Bahia, Salvador 40110-902, Brazil

**Keywords:** *Candida maltosa*, *Candida norvegica*, *Kazachstania telluris*, *Schizosaccharomyces pombe*, *Scheffersomyces insectosus*, *Meyerozyma guilliermondii*, *Brettanomyces bruxellensis*, *Kazachstania exigua*, *Starmerella lactis-condensi*

## Abstract

(1) Background: The identification of microorganisms includes traditional biochemical methods, molecular biology methods evaluating the conserved regions of rRNA, and the molecular biology of proteins (proteomics), such as MALDI-TOF MS mass spectrometry. This work aimed to identify the biodiversity of yeasts associated with stingless bee species’ honey and pollen, *Melipona scutellaris*, *Nannotrigona testaceicornes*, and *Tetragonisca angustula*, from the region of São Gonçalo dos Campos-Bahia (BA) state, Brazil. (2) Methods: Cellular proteins were extracted from 2837 microbial isolates (pollen and honey) and identified via MALDI-TOF MS. The identified yeast species were also compared to the mass spectra of taxonomically well-characterized reference strains, available from the National Center of Biotechnology Information (NCBI) database. (3) Results: Nine yeast species were identified: *Candida maltosa*, *Candida norvegica*, *Kazachstania telluris*, *Schizosaccharomyces pombe*, *Scheffersomyces insectosus*, *Meyerozyma guilliermondii*, *Brettanomyces bruxellensis*, *Kazachstania exigua*, and *Starmerella lactis-condensi. Nannotrigona testaceicornes* pollen had the highest number of yeast colonies. The yeasts *Brettanomyces bruxellensis* and *Kazachstania telluris* showed high populations in the samples of *Nannotrigona testaceicornes* and *Melipona scutellaris*, respectively. This work shows that there is some sharing of the same species of yeast between honey and pollen from the same beehive. (4) Conclusions: A total of 71.84% of the identified species present a high level of confidence at the species level. Eight yeast species (*Candida maltosa*, *Candida norvegica*, *Kazachstania telluris*, *Schizosaccharomyces pombe*, *Scheffersomyces insectosus*, *Meyerozyma guilliermondii*, *Kazachstania exigua*, and *Starmerella lactis-condensi*) were found for the first time in the samples that the authors inspected. This contributes to the construction of new knowledge about the diversity of yeasts associated with stingless bee products, as well as to the possibility of the biotechnological application of some yeast species.

## 1. Introduction

Bees belonging to the *Hymenoptera* order and the *Apidae* family, *Apinae* subfamily, *Meliponini* tribe, are also popularly known as stingless bees, because they have atrophied aculei and, therefore, are unable to sting. The group includes more than 500 species and 61 genera known worldwide, distributed in tropical and subtropical regions (tropical America, Africa, Southeast Asia, and Australia) [1,2,3,4].

Meliponines form perennial colonies where they store food and keep their offspring protected. The stingless bee colony is built with different materials; some of them are taken from nature, such as clay, propolis, and resins, and others are produced or processed inside the colony, such as wax, cerumen, and geopropolis. The colony consists of two main elements: the nest and the food pots [3]

Pollen and nectar are important resources for the development and maintenance of the offspring and colony growth. Pollen is a natural source of proteins, lipids, minerals, and vitamins and is also a rich source of phytochemicals (flavonoids and phenolic acids) [5,6]. Nectar is the main source of carbohydrates (sucrose, glucose, and fructose) used as a source of energy by bees [7,8].

Pollen is collected from flowers through the corbicula and is taken to the colony. This pollen is stored in wax pots, where it is processed by bees that deposit nectar and some secretions rich in enzymes, together with microorganisms. Afterward, the pots are sealed with wax, making the environment conducive to microbial fermentation that transforms it into “samburá” or “pot pollen” rich in essential nutrients for the survival of the colony [9].

The nectar of the flowers is transported through the melliferous vesicle, where it receives some enzymes (salivary secretions from the glands in the abdomen and enzymes from the cephalic glands). The nectar begins to be processed, and later, inside the colony, this product is matured and transformed into honey [9,10,11]. The honey produced by meliponines has very different sensory, physical, and chemical characteristics when compared to that produced by Apis bees; they differ in color, flavor, and viscosity, have a high moisture content, lower total carbohydrate content, are more acidic, and their crystallization is slower. It also has a wider variety of microorganisms (bacteria and yeast) that induce fermentation [8,12,13,14,15].

The microbiota associated with stingless bees are very diverse, including bacteria, fungi, and yeasts. Microorganisms can interact with bees by symbiosis; contribute to bee nutrition; produce biomolecules that help transform substances such as nectar and pollen; and break down molecules that the bee is unable to digest. Some species can act by producing toxins and/or antimicrobial compounds that have the purpose of inhibiting the growth of pathogens [16,17,18,19]. Yeasts associated with stingless bees can be found in plants and flowers (pollen and nectar) [20], as well as in the bees’ digestive tract (secretions), in fermented pollen (samburá), in the honey and propolis produced by these bees, and even in their larvae [21].

The use of yeast in biotechnological processes has been feasible, as they have low pathogenic potential, are easy to handle and apply, present rapid cell multiplication, are potential fermenters and enzyme producers, and adapt well to different substrates [22].

The microbiota associated with pollen and honey from stingless bees are still poorly studied. The biodiversity of yeasts present in stingless bee honey is also not well-known. The anaerobic environment, high humidity, and concentration of fermentable sugars in the pots (closed with cerumen) have favored the interactions and development of a variety of species of microorganisms, mainly yeasts. This environment has a high concentration of sugars subject to the microbial fermentative process [16].

The main methods used to identify microorganisms include traditional biochemical methods, molecular biology methods evaluating the conserved regions of rRNA (ribosomal RNA), and the molecular biology of proteins (proteomics), such as mass spectrometry by MALDI-TOF MS. The matrix-assisted laser desorption/ionization time-of-flight mass spectrometry and time-of-flight (MALDI-TOF MS) analyzers evaluate proteins obtained from microbial cells which are specific for each species; this technique has demonstrated high potential in the identification of yeasts [23,24,25]. In addition, MALDI-TOF is a scientifically up-to-date technique used in several studies for the microbial identification of food samples [22,26,27,28,29,30,31].

Given the above, this work aimed to isolate and identify the biodiversity of cultivable yeasts associated with the honey and pollen of the stingless bee species, *Melipona scutellaris*, *Nannotrigona testaceicornes*, and *Tetragonisca angustula*, from the region of São Gonçalo dos Campos-BA, Brazil. In addition to contributing to new knowledge about the diversity of yeast in stingless bees, this study also presents the discovery of new species of yeast that are of biotechnological interest.

## 2. Materials and Methods

### 2.1. Sample Collection

Honey and pollen samples (10 samples containing 20 mL of honey and 15 g of pollen from each beehive analyzed) of the species *Melipona scutellaris* (Uruçu), *Nannotrigona testaceicornes* (Iraí), and *Tetragonisca angustula* (Jataí) were obtained from a meliponary located in the municipality of São Gonçalo dos Campos-BA, Brazil (12°25′38″ S; 38°58′26″ W; altitude 233 m), in September 2019 (Figure 1).

Honey samples (20 mL) were collected using sterile disposable syringes, and pollen samples (15 g) using a sterile spatula. Samples were collected in triplicate, directly from the food pots located inside the hives. The samples were stored in sterile 50 mL Falcon tubes, transported in isothermal boxes, stored at 4 °C, and then taken to the Microbiology Laboratory, located at the Faculty of Pharmacy of the Federal University of Bahia—UFBA to carry out the yeast isolation. Species identification using the MALDI-TOF MS technique was carried out at the Biology Department, Microbiology Sector, at the Federal University of Lavras—UFLA, in Minas Gerais.

### 2.2. Dilution, Plating, and Isolation

The samples were diluted in the proportion of 5 g of sample to 45 mL of sterile 0.1% peptone water (first dilution 10^−1^). From this dilution, successive dilutions were obtained until the 10^−3^ dilution, aseptically transferring 1 mL of the starting solutions (10^−1^) to a tube containing 9 mL of the diluent solution (10^−2^). Then, 1 mL of the 10^−2^ dilution was transferred to a new tube containing 9 mL of the diluent solution (10^−3^) [32].

Samples were plated (100 µL or 0.1 mL) in three different culture media. The samples were plated in triplicate, following the spreading technique, inoculating 0.1 mL of the dilutions of each sample in three different culture media, Peptone Dextrose Yeast Extract Agar (YEPD—agar 15 g/L, bacteriological peptone 20 g/L, dextrose 20 g/L, Yeast extract 10 g/L; Sigma, St. Louis, MO, USA), Malt Extract Agar (YM—agar 15 g/L, bacteriological peptone 5 g/L, malt 30 g/L; Acumedia, St. Louis, MO, USA), and Sabouraud Dextrose Agar (SDA—agar 15 g/L, bacteriological peptone 10 g/L, dextrose 40 g/L; Merck, Düren, Germany). All culture media were supplemented with ampicillin (10 mg/L; Neo Química, São Paulo, SP, Brazil) [33].

Subsequently, the plates were incubated at 28 °C for five days and after growth, the colonies that showed typical characteristics were counted. The results of counts were expressed in log CFU/g^−1^. The count was performed by similarity of typical yeast colonies. Subsequently, the morphotypes were identified by the square root (√) of each morphotype.

Colonies were selected based on morphological characteristics: colony size, border type and structure, color, texture, appearance, elevation, brightness, and shape. The different morphotypes were isolated and purified through successive replications using the depletion technique (successive replication by compound streaks), transferring the selected colony to another plate containing their respective culture media, incubating at 28 °C for 48 h [34].

### 2.3. Preservation and Reactivation of Yeasts

The yeast isolates were transferred through a platinum loop to cryotubes (Kasvi, São Paulo, SP, Brazil) containing 800 µL of the specific liquid culture medium (YEPD, YM, SDA, described in Section 2.2). The tubes were incubated at 28 °C for 24 h, after which 200 µL of glycerol (Sigma, St. Louis, MO, USA) was added, and they were again incubated at 4 °C for 24 h to adapt the microorganism to low temperatures. The tubes were stored in a freezer at −20 °C [22,35].

The Isolates preserved at −20 °C were reactivated, transferring an aliquot through a platinum loop to tubes containing the respective broths, and after incubation at 28 °C for 48 h, the samples were seeded in plates containing the culture media described and were incubated again for colony growth. This procedure was important for the efficiency of protein extraction (MALDI-TOF MS analysis).

### 2.4. Protein Extraction

The isolated yeast colonies were transferred separately, in triplicate, through a 10 µL platinum loop to a 1.5 mL tube (Eppendorf) (Kasvi, São Paulo, SP, Brazil) containing 300 μL of deionized water, generating a homogeneous suspension of the transferred cells; then, the tubes were vortexed for about 30 s. Subsequently, 900 µL of absolute ethanol (99.8% (*v*/*v*); Sigma, St. Louis, MO, USA) was added to this suspension for microbial inactivation (cell lysis). The tubes were again vortexed (Quimis, São Paulo, SP, Brazil) for 30 s and centrifuged for 2 min at 13,000 revolutions per minute (RPM). The supernatant was discarded using a pipette. The centrifugation step and removal of the supernatant were repeated for complete alcohol removal [36,37].

Cellular proteins were extracted following the extraction protocol (Bruker Flex Control 3.4, Bruker Daltonics Inc., Billerica, MA, USA), adding 50 μL of formic acid at 70% (*v*/*v*) (300 μL of water/700 μL of formic acid (Sigma, St. Louis, MO, USA) 100% (*v*/*v*)) to the pellet and, after homogenization, adding and removing with the aid of a pipette; the suspension was vortexed. Subsequently, 50 μL of acetonitrile (Sigma, St. Louis, MO, USA) was added to the suspension. After a new centrifugation at 13,000 RPM for 2 min, the supernatant was forwarded for analysis [38].

### 2.5. Identification of Yeast Species by MALDI-TOF MS

A total of 1 µL of the supernatant was applied to the 96-well MALDI-TOF MS target plate (Bruker Flex Control 3.4, Bruker Daltonics Inc., Billerica, MA, USA), in triplicate, and allowed to dry in the environment. The sample was covered with 1 µL of the matrix (saturated solution of α-cyano-4-hydroxy-cinnamic acid (Sigma, St. Louis, MO, USA) in 50% acetonitrile (Sigma, St. Louis, MO, USA) and 2.5% trifluoroacetic acid (Merck, Düren, Germany)), and after drying, the plate was inserted into the equipment (MALDI-TOF MS) to obtain the spectra (sequence of peaks) [38].

#### 2.5.1. Acquisition of Mass Spectra

The “Bruker Flex Control 3.4, Bruker Daltonics Inc., Billerica, MA, USA” software was used to acquire protein profiles of isolated yeast strains. Samples were analyzed in triplicate. The acquired spectral profiles were then analyzed and processed using the “Biotyper RTC 3—Bruker Daltonics Inc., Billerica, MA, USA” software.

#### 2.5.2. Statistical/Molecular Analysis of Mass Spectra

All mass spectra derived from MALDI-TOF MS analysis were retrieved as text files (txt format) and imported into BioNumerics v. 7.10 (Applied Maths, Ghent, Belgium) for further analysis. Mass fingerprinting for each isolate was recorded according to the spectrum qualitatively best in terms of peak intensity. Similar spectra were determined by Pearson’s correlation coefficient, which grouped spectra with those taxonomically similar to the reference strains using unweighted paired group with arithmetic mean (UPGMA). Isolates were considered identified when their spectra were grouped with reference taxonomic strains. The grouping of yeast species identified by MALDI-TOF MS was carried out and the cluster related to the grouping was constructed.

The identified yeast species were also compared to the mass spectra of taxonomically well-characterized reference strains, available at the National Center of Biotech Information (NCBI) database (https://www.ncbi.nlm.nih.gov/taxonomy accessed on 16 February 2024). The yeast species identified in this study received a taxonomic update according to https://www.mycobank.org/ (accessed on 16 February 2024).

## 3. Results

### 3.1. Yeast Count

Two thousand eight hundred and thirty-seven (2837) yeast colonies were isolated, obtained from the pollen and honey of *Melipona scutellaris*, *Nannotrigona testaceicornes*, and *Tetragonisca angustula* bees, identifying a total of nine different yeast species. The isolates were identified by the MALDI-TOF MS technique. Figure 2 and Figure 3 show the yeast count (log CFU/g^−1^).

In the present study, nine yeast species (*Candida maltosea*, *Candida norvegica*, *Kazachstania telluris*, *Schizosaccharomyces pombe*, *Scheffersomyces insectosus*, *Meyerozyma guilliermondii*, *Brettanomyces bruxellensis*, *Kazachstania exigua*, *Starmerella lactis-condensi)* were identified in the pollen of *Nannotrigona testaceicornes* and *Tetragonisca angustula* bees (Figure 2), and seven yeast species (*Kazachstania telluris*, *Schizosaccharomyces pombe*, *Scheffersomyces insectosus*, *Meyerozyma guilliermondii*, *Brettanomyces bruxellensis*, *Kazachstania exigua*, *Starmerella lactis-condensi*) were found in the honey of the three studied stingless bee species (Figure 3).

Figure 2 shows that the highest number of yeasts was found in the pollen of *Nannotrigona testaceicornes*, which obtained the highest number of yeast colonies, with the predominant species *Kazachstania exigua* (5.0 log CFU/g^−1^) and *Starmerella lactis-condensi* (4.5 log CFU/g^−1^); other species such as *Brettanomyces bruxellensis* (4.0 log CFU/g^−1^) and *Meyerozyma guilliermondii* (3.0 log CFU/g^−1^) were also found. In *Tetragonisca angustula* pollen, the growth of *Candida norvegica* (3.5 log CFU/g^−1^), *Candida maltosa* (3.0 log CFU/g^−1^), *Kazachstania telluris* (3.0 log CFU/g^−1^), *Schizosaccharomyces pombe* (2.5 log CFU/g^−1^), and *Scheffersomyces insectosus* (1.5 log CFU/g^−1^) was observed; the pollen of these bees showed a greater diversity of yeasts. *Melipona scutellaris* samples did not show yeast growth.

In Figure 3, the yeast species *Kazachstania telluris* (8.5 log CFU/g^−1^) and *Schizosaccharomyces pombe* (8.5 log CFU/g^−1^) showed greater growth in *Tetragonisca angustula* honey. The yeasts *Brettanomyces bruxellensis* (8.0 log CFU/g^−1^) and *Kazachstania telluris* also showed high values in samples from *Nannotrigona testaceicornes* and *Melipona scutellaris*, respectively. *Kazachstania exigua* (7.0 log CFU/g^−1^), *Starmerella lactis-condensi* (6.0 log CFU/g^−1^), and *Meyerozyma guilliermondii* (5.0 log CFU/g^−1^) were also found in honey from *Nannotrigona testaceicornes*, showing greater diversity. The yeasts *Schizosaccharomyces pombe* (5.0 log CFU/g^−1^) and *Brettanomyces bruxellensis* (4.0 log CFU/g^−1^) were also observed in *Melipona scutellaris* honey.

The present work shows that there is a sharing of the same species of yeast between honey and pollen from the same hive. This can be seen by the presence of yeasts *Kazachstania exigua*, *Starmerella lactis-condensi*, *Brettanomyces bruxellensis*, and *Meyerozyma guilliermondii* in honey and pollen of *Nannotrigona testaceicornes*, as well as the presence of yeasts *Scheffersomyces insectosus*, *Kazachstania telluris*, and *Schizosaccharomyces pombe* in honey and pollen of the *Tetragonisca angustula* bee.

The sharing of yeasts between different bee species was also observed. The yeasts *Kazachstania telluris* and *Schizosaccharomyces pombe* were found in samples of *Tetragonisca angustula* and in honey from *Melipona scutellaris*. *Brettanomyces bruxellensis* was the only yeast species associated with the three bee species.

Already in exclusive association, *Kazachstania exigua*, *Starmerella lactis-condensi*, and *Meyerozyma guilliermondii* were found in honey and pollen of *Nannotrigona testaceicornes*, and *Candida norvegica* and *Candida maltosa* in *Tetragonisca angustula* pollen.

### 3.2. Yeast Proteomic Identification Using MALDI-TOF MS/Genbank

Analysis of yeast isolates by MALDI-TOF MS provided typist scores inside the National Center of Biotech Information (NCBI—https://www.ncbi.nlm.nih.gov/taxonomy accessed on 16 February 2024) algorithm rotation range from 2053 to 2892. This refers to the reliability of microbial identification at the species level. Among the identified isolates, 2038 (71.84%) and 799 (28.16%) obtained scores of 2300–3000 (+++) and 2000–2299 (++), respectively. No yeast colonies were found for *Melipona scutellaris* pollen (Table 1).

The scores obtained by the MALDI-TOF MS identification technique indicate that in the range from 2300 to 3000, the confidence at the species level is highly probable, represented by the symbol (+++); the range from 2000 to 2299 indicates highly probable genus level and probable identification at the species level (++); the range from 1700 to 1999 indicates probable identification at the gender level (+); and the range from 1700 to 0 indicates unreliable identification (−) (Silva et al. 2021). Therefore, the yeast isolates identified in this study have a high level of confidence at the species level (Table 1).

Spectra generated by MALDI-TOF MS were grouped for similarity by Pearson’s correlation coefficient, which grouped the spectra with those taxonomically similar to the reference strains using unweighted paired group with arithmetic mean (UPGMA—BioNumerics software v. 7.10).

## 4. Discussion

The association between yeasts and stingless bees was first reported in the work of Rosa et al. [21]. Yeasts associated with stingless bees can come from plants and flowers, pollen and nectar collected by them, honey, fermented pollen, brood cells, larvae, and the bees’ digestive tract [21].

Yeasts play an important role in the processing and transformation of pollen inside the colonies, as they secrete enzymes that participate in biochemical processes and contribute to the transformation of bee pollen into fermented pollen, making nutrients bioavailable and improving the food nutritional quality. Yeasts also act in symbiotic relationships (yeast–insect), in the synthesis of nutrients such as amino acids, vitamins, lipids, and pheromones, in addition to being protective by degrading toxic compounds [39,40].

Rosa et al. [21], Teixeira et al. [41], and Daniel et al. [42] identified the genera *Candida*, *Hyphopichia*, *Kodamaea*, *Pichia*, *Wickerhamiella*, and *Zygosaccharomyces* in the pollen of stingless bees, and the species *Starmerella neotropicalis*, *Starmerella meliponinorum*, *Starmerella apicola*, *Kodamaea ohmeri*, and *Wickerhamiella versalitis* were also found in the pollen collected by stingless bees.

Although honey has physical–chemical properties that prevent the development and proliferation of microorganisms, such as low water activity, low pH, high concentration of sugars, high osmolarity, in addition to the presence of antimicrobial compounds, it is believed that the inoculation of yeasts in honey comes from the bees themselves [43,44].

Normative Instruction No. 11/2000 [45], which establishes quality standards for *Apis mellifera* honey, does not require microbiological analysis for these products; it only refers to the hygienic–sanitary standards and good manufacturing practices for establishments that prepare/manufacture food. In addition, Normative Instruction No. 161, also of 1 July 2022 [46], which establishes the food microbiological standards, also does not require microbiological control, and excludes the need for analysis of pathogens such as *Listeria monocytogenes* for honey. The Resolution of the Collegiate Board of ANVISA No. 724, of 1 July 2022 [47], regulates the microbiological quality control of foods in force in Brazil.

The following species of yeast have already been described in honey from stingless bees: *Starmerella meliponinorum*, *Metschnikowia* sp., *Zygosaccharomyces rouxii*, *Zygosaccharomyces mellis*, *Zygosaccharomyces bailii*, *Saccharomyces cerevisiae*, *Saccharomyces mellis*, *Saccharomyces rosei*, *Lachancea fermentati*, *Pichia anomala*, *Pichia kudriavze Wickerhamomyces anomalus*, *Dekkera bruxellensis*/*Brettanomyces bruxellensis*, and *Kloeckera africana* [21,48,49]. Studies performed by Rosa et al. [21], Fernandes et al. [50], and Barbosa et al. [51] present average yeast values for honey from stingless bees: *Tetragonisca angustula*, *Melipona quadrifasciata*, and *Frieseomelitta* (4.41 log CFU/g^−1^), *Melipona mandaçaia* (3.13 log CFU/g^−1^), *Melipona asilvai* (1.70 log CFU/g^−1^), *Partamona* sp. (2.70 log CFU/g^−1^), *Scaptotrigona* sp. (3.07 log CFU/g^−1^), and *Melipona fasciculata* (2.02 log CFU/g^−1^).

In this study, it was observed that the yeast *Brettanomyces bruxellensis* is associated with honey from *Nannotrigona testaceicornes* and *Melipona scutellaris* bees, and pollen from *Nannotrigona testaceicornes* bees. This result corroborates the study carried out by Barbosa et al. [51], who identified the species *Brettanomyces bruxellensis/Dekkera bruxellensis* in samples of honey from bees *Melipona mandaçaia* and *Melipona asilvai* that live in the Brazilian tropical dry forest.

*Brettanomyces bruxellensis* yeast is capable of developing in a wide temperature range (19 to 35 °C), with 25 to 28 °C being the ideal temperature range for its development; it metabolizes carbon sources such as glucose, fructose, sucrose, maltose, and galactose; it uses ammonia, proline, arginine, and nitrate as nitrogen sources; moreover, it is ethanol-tolerant; it resists low pH; it tolerates sulfur dioxide; it produces metabolites such as acetic acid and volatile phenols (4-vinylphenol, 4-vinylguaiacol, 4-ethylphenol and 4-ethylguaiacol) [52,53,54].

The greater industrial relevance of this yeast *Brettanomyces bruxellensis* is highlighted due to the fact that it is considered a deteriorating microorganism for wine; this is due to the production of volatile phenols such as 4-vinylphenol, 4-vinylguaiacol, 4-ethylphenol, and 4-ethylguaiacol, which gives aroma and unpleasant taste to the product. Despite being considered a problem for wine production, *Brettanomyces bruxellensis* has great technological potential for the manufacture of complex spontaneous fermentation beers, playing an important role for the organoleptic characteristics of sour beers, such as the Belgian Lambics and Gueuze, the American Coolship ales, and the German Berliner Weisse [54,55].

*Brettanomyces bruxellensis* can produce a range of higher alcohols and several esters that contribute to a more complex sensory perception. The presence of 4-ethyl-phenol and 4-ethyl-guaiacol produced by this species of yeast is considered interesting when spicy notes are desired in beer. In addition to wine and beers with complex spontaneous fermentation, *Brettanomyces bruxellensis* has also been isolated from products such as kombucha, kefir, tequila, cider, and olives [56,57,58,59,60,61].

Eight yeast species (*Candida maltosa*, *Candida norvegica*, *Kazachstania telluris*, *Schizosaccharomyces pombe*, *Scheffersomyces insectosus*, *Meyerozyma guilliermondii*, *Kazachstania exigua*, *Starmerella lactis-condensi*) were found for the first time in the samples that the authors inspected. This contributes to the construction of new knowledge about yeast diversity associated with honey and pollen from stingless bees, as well as the possibility of biotechnological application of some species.

*Schizosaccharomyces pombe* was found in honey samples from *Tetragonisca angustula* and *Melipona scutellari* bees, and in *Tetragonisca angustula* pollen. Benito et al. [62] identified the presence of *Schizosaccharomyces pombe* in the honey and honeycomb of *Apis mellifera* bees, suggesting honey as the primary habitat for this yeast. The affinity for this product is due to the variability of moisture content and water activity during maturation, allowing the propagation of this yeast [63].

The yeast of the genus *Schizosaccharomyces* began to be studied, due to its capacity of biological degradation of malic acid. The first works involved the selection of these yeasts that had good alcohol yield and high consumption of malic acid, without developing negative organoleptic characteristics [64].

*Schizosaccharomyces pombe* is very tolerant to ethanol and sulfur dioxide concentrations, tolerates low pH values, low aw (water activity), low temperatures, has no nutritional requirements, is resistant to high sugar concentrations, and has a good growth capacity [62,65]. This yeast species has also been strongly linked to its ability to deacidify wine. Wine acidity is a fundamental parameter for the organoleptic balance and for better quality and microbiological stability of the must and/or product. This parameter essentially depends on the concentration of tartaric acid and malic acid [66,67].

The study carried out by Benito et al. [62] identified three genetically different strains of *Schizosaccharomyces pombe* that showed properties of enological interest. These strains were compared to the *Saccharomyces cerevisiae* standard. *Schizosaccharomyces pombe* showed good results in terms of malic acid deacidification, significantly reducing the levels of biogenic amines and ethyl carbamate precursors, without the need for the secondary action of bacteria for malolactic fermentation. This yeast species may become an important microorganism with different biotechnological applications.

*Candida norvegica* was found in the pollen of *Tetragonisca angustula* bees. This yeast has been associated with the fermentation process of Negrinha de Freixo [68] green table olives. A study carried out by Oliveira et al. [69] showed the potential use of *Candida norvegica* as a producer of the β-glucosidase enzyme, as well as the antimicrobial activity against the action of the pathogenic yeast *Cryptococcus neoformans*. This activity may be related to the capacity of the yeast to produce a substance known as mycocin, which has antifungal properties. *Starmerella lactis-condensi* was found in honey and in the pollen of *Nannotrigona testaceicornes* bees. A study carried out by Lachance et al. [70] proves that *Starmerella lactis-condensi* is associated with insects such as beetles and bees that visit ephemeral flowers. It is capable of fermenting most sugars but prefers fructose as a carbon source. Despite giving wines an intense floral flavor, *Starmerella lactis-condensi* is also frequently associated with wine spoilage [71,72,73].

*Starmerella lactis-condensi* is osmotolerant, making possible its development in beverages with high sugar content, concentrated fruit juices, confectionery products, honey, dried fruits, and jellies [74,75,76,77,78,79]. This yeast is considered important in the fermentation of traditional balsamic vinegar that is produced with cooked grape must, a selective and stressful medium for the growth of yeasts, due to its high sugar content and low values of pH [80].

*Meyerozyma guilliermondii* is associated with *Nannotrigona testaceicornes* pollen [81,82,83,84,85]. *Meyerozyma guilliermondii* is widely distributed in soil, plants, sea water, and processed foods, in addition to the mucosa. This yeast has been frequently seen as an opportunistic pathogenic agent because it is resistant to several antifungal agents. Despite being studied due to its clinical importance, *Meyerozyma guilliermondii* is considered of interest for the biotechnological area. They are promising in the production of enzymes such as inulinase, in the production of biofuel, flavors, and riboflavins, as well as in the fermentation of xylose [81,82,83,84,85].

The *Scheffersomyces insectosus* isolated from the pollen and honey of the bee *Tetragonisca angustula* in this study has also been isolated from wood and sugarcane bagasse [86,87]. This material is considered lignocellulosic, composed of cellulose, hemicellulose, and lignin, hemicellulose being considered attractive for the production of biofuels. The decomposition of hemicellulose results in a solution of sugars, mainly xylose and glucose; for fermentation of these hydrolysates, it is necessary to use yeast capable of transforming both types of sugar. Although *Saccharomyces cerevisiae* yeast is capable of fermenting glucose, it is unable to efficiently ferment xylose, with the role of *Scheffersomyces insectosus* being important, which can produce ethanol through the fermentation of this sugar [86,87]. 

*Candida maltosa* found in *Tetragonisca angustula* bee pollen is an unconventional yeast that is not commonly used in fermentation processes but can be used for cellular protein production. It is capable of growing on various carbon sources and can degrade hydrocarbons such as phenylalkanes and n-alkanes. *Candida maltosa* is also used in bioremediation processes. It is used for the biodegradation of oils in an aqueous medium [88,89]. *Candida maltose* has also been isolated from fermentation vats for the production of bioethanol.

The genus *Kazachstania* is closest to the genus *Saccharomyces*, which are known to completely ferment all sugars and are capable of producing low amounts of undesirable by-products such as acetic acid. *Kazachstania* can be found in diverse habitats such as soil, animals, water, and fermented products [90,91,92,93].

*Kazachstania* spp. species are the most frequently cited in studies. These yeasts have already been isolated from natural habitats such as soil, mangroves, and fruits and fermented products such as grape must, kefir, and dairy products [90]. Studies carried out by Perez et al. [94] showed that *Kazachstania exigua* has inhibitory activity against the pathogenic bacteria *Pseudomonas aeruginosa*, *Escherichia coli*, and *Klebsiella pneumoniae*. *Kazachstania telluris* species are glucose and lactic acid fermenters; however, they can be considered pathogenic because they cause infections in animals [93,95,96]. The yeast *Kazachstania exigua* showed growth in pollen and honey samples of *Nannotrigona testaceicornes*; the yeast *Kazachstania telluris* grew in the pollen and honey samples of *Tetragonisca angustula*, as well as in the honey of *Melipona scutellaris*. These species adapted well to the three-culture media used.

Spectra generated by MALDI-TOF MS were grouped for similarity by Pearson’s correlation coefficient (Figure 4). Thus, it is possible to observe the grouping between yeast species of the same microbial genus (*Kazachstania* spp. and *Candida* spp.). However, the species *Scheffersomyces insectosus* and *Meyerozyma guilliermondii* were grouped into two distinct groups. This may have happened due to the differentiation of the protein constitution of these strains, caused by different factors, from the original environment to small genetic mutations [52].

## 5. Conclusions

We know that there are few studies related to the identification of yeast species in different honeys and pollen of stingless bees belonging to the State of Bahia, Brazil, an important center for the cultivation of stingless bees and the production/marketing of this type of honey. The species cataloged in this work were *Brettanomyces bruxellensis*, *Candida maltosa*, *Candida norvegica*, *Kazachstania telluris*, *Schizosaccharomyces pombe*, *Scheffersomyces insectosus*, *Meyerozyma guilliermondii*, *Kazachstania exigua*, and *Starmerella lactis-condensi*. The species *Candida maltosa*, *Candida norvegica*, *Kazachstania telluris*, *Schizosaccharomyces pombe*, *Scheffersomyces insectosus*, *Meyerozyma guilliermondii*, *Kazachstania exigua*, and *Starmerella lactis-condensi*, found in this study, are unprecedented in honey and stingless bee pollen and can contribute to the construction of new knowledge about the diversity of yeasts associated with these substances. This also directly enables the biotechnological application of these yeast species isolated and identified to the species level. The identification of new species highlights the potential of substrates for yeast development, in addition to the importance of studies on the biodiversity of yeasts associated with honey and stingless bee pollen.

Protein molecular analysis using the MALDI-TOF MS technique offers excellent taxonomic resolution for identifying yeasts, including closely related species, with foods of different origins. In addition, MALDI-TOF MS has the advantage of low cost of identification of microbial species, ease of operation, and short response time. The yeast species identified in this study received a taxonomic update according to https://www.mycobank.org/. The development of referential databases of industrial interest is fundamental for its further application in practice. More studies are underway to examine the relationship between physicochemical and sensory characteristics of honey and stingless bee pollen with the dynamics of isolated and identified yeasts. However, the potential of employing MALDI-TOF MS to monitor the yeast diversity of honeys and pollen of stingless bees of the species *Melipona scutellaris*, *Nannotrigona testaceicornes*, and *Tetragonisca angustula* from the region of São Gonçalo dos Campos in the State of Bahia is defended in our study.

## 6. Future Perspectives

The scientific interest in investigating the yeast composition of stingless bees’ honey/pollen is a slow cumulative effort to contribute to the scientific community for research on Brazilian meliponines. The contribution is also important for the understanding of beekeeping processes and for final product quality purposes, which will eventually support stingless beekeeping, agriculture, and Brazilian consumers. A microbiological standard norm for Brazilian honey produced by stingless bees is necessary to establish a quality standard for this important agricultural food. Our study presents an important path in the stingless bees’ honey microbiological classification in Brazil. Future studies for complete microbial biodiversity (eukaryotes and prokaryotes) are necessary. Based on our studies, the MALDI-TOF MS proteomic technique is promising for this purpose, allowing quality microbial identification at the species level, in addition to providing new microbial isolates for different biotechnological applications, spanning food and even medical areas.

## Figures and Tables

**Figure 1 microorganisms-12-00678-f001:**
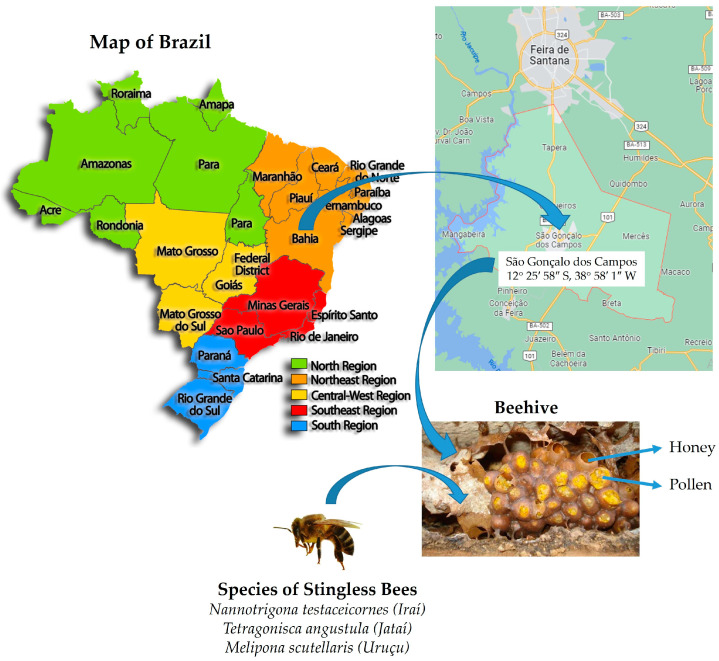
Location for the collection of evaluated honey and pollen samples.

**Figure 2 microorganisms-12-00678-f002:**
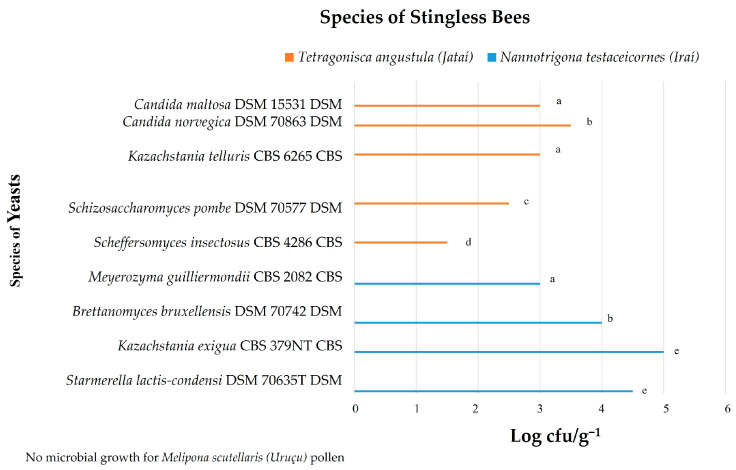
Yeast identified in the beehive pollen from studied stingless bees. Equal letters indicate that there is no statistical difference among the microbial counts, according to the Tukey test.

**Figure 3 microorganisms-12-00678-f003:**
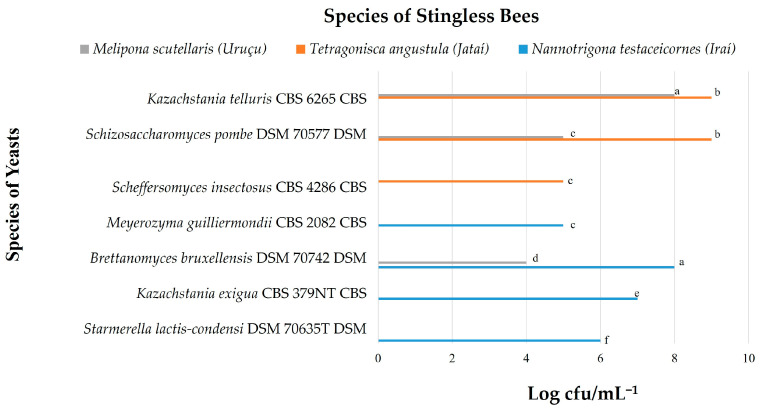
Yeasts identified in the honey from studied stingless bees. Equal letters indicate that there is no statistical difference among the microbial counts, according to the Tukey test.

**Figure 4 microorganisms-12-00678-f004:**
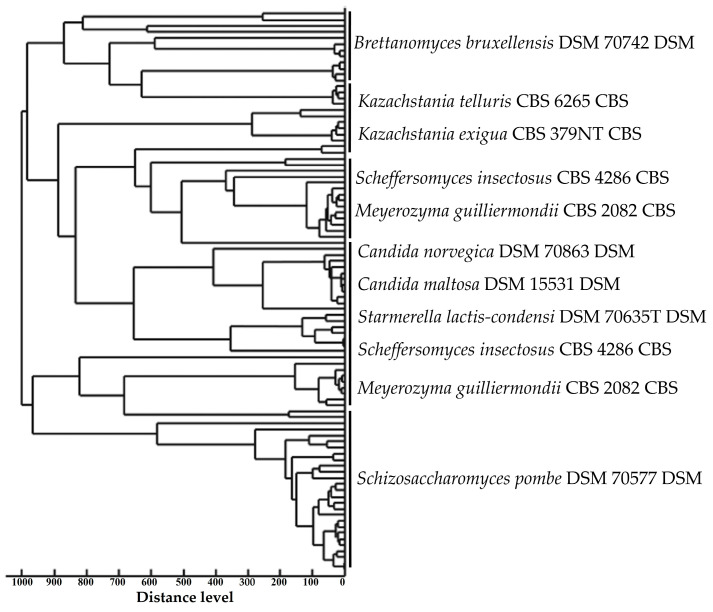
Taxonomically similar MALDI-TOF MS spectra.

**Table 1 microorganisms-12-00678-t001:** Diversity of yeast species identified in honey/pollen samples according to different culture media using the Maldi-Tof MS/Genbank technique.

Morphotype Code (Number of Isolates)	Sample	Stingless Bee Species (“Popular Name” in Brazil)	Culture Medium	Yeast Species	NCBI Identifier	Score Value	Symbol
1 (230)	POLLEN	*Nannotrigona testaceicornes* (Iraí)	SABOURAUD	*Starmerella lactis-condensi* DSM 70635T DSM	NCBI:txid 45562	2091	(++)
2 (142)	POLLEN	*Nannotrigona testaceicornes* (Iraí)	SABOURAUD	*Starmerella lactis-condensi* DSM 70635T DSM	NCBI:txid 45562	2091	(++)
3 (137)	POLLEN	*Nannotrigona testaceicornes* (Iraí)	SABOURAUD	*Kazachstania exigua* CBS 379NT CBS	NCBI:txid 34358	2522	(+++)
4 (122)	POLLEN	*Nannotrigona testaceicornes* (Iraí)	YEAST EXTRACT	*Brettanomyces bruxellensis* DSM 70742 DSM	NCBI:txid 5007	2546	(+++)
5 (99)	POLLEN	*Nannotrigona testaceicornes* (Iraí)	YEAST EXTRACT	*Meyerozyma guilliermondii* CBS 2082 CBS	NCBI:txid 4929	2346	(+++)
6 (87)	POLLEN	*Nannotrigona testaceicornes* (Iraí)	YEAST EXTRACT	*Brettanomyces bruxellensis* DSM 70742 DSM	NCBI:txid 5007	2546	(+++)
7 (230)	POLLEN	*Nannotrigona testaceicornes* (Iraí)	MALT EXTRACT	*Kazachstania exigua* CBS 379NT CBS	NCBI:txid 34358	2522	(+++)
8 (205)	HONEY	*Nannotrigona testaceicornes* (Iraí)	SABOURAUD	*Brettanomyces bruxellensis* DSM 70742 DSM	NCBI:txid 5007	2546	(+++)
9 (56)	HONEY	*Nannotrigona testaceicornes* (Iraí)	SABOURAUD	*Kazachstania exigua* CBS 379NT CBS	NCBI:txid 34358	2091	(++)
10 (73)	HONEY	*Nannotrigona testaceicornes* (Iraí)	YEAST EXTRACT	*Brettanomyces bruxellensis* DSM 70742 DSM	NCBI:txid 5007	2642	(+++)
11 (34)	HONEY	*Nannotrigona testaceicornes* (Iraí)	YEAST EXTRACT	*Brettanomyces bruxellensis* DSM 70742 DSM	NCBI:txid 5007	2702	(+++)
12 (210)	HONEY	*Nannotrigona testaceicornes* (Iraí)	YEAST EXTRACT	*Starmerella lactis-condensi* DSM 70635T DSM	NCBI:txid 45562	2057	(++)
13 (82)	HONEY	*Nannotrigona testaceicornes* (Iraí)	YEAST EXTRACT	*Kazachstania exigua* CBS 379NT CBS	NCBI:txid 34358	2836	(+++)
14 (57)	HONEY	*Nannotrigona testaceicornes* (Iraí)	MALT EXTRACT	*Kazachstania exigua* CBS 379NT CBS	NCBI:txid 34358	2088	(++)
15 (32)	HONEY	*Nannotrigona testaceicornes* (Iraí)	MALT EXTRACT	*Kazachstania exigua* CBS 379NT CBS	NCBI:txid 34358	2736	(+++)
16 (47)	POLLEN	*Tetragonisca angustula* (Jataí)	SABOURAUD	*Scheffersomyces insectosus* CBS 4286 CBS	NCBI:txid 45590	2342	(+++)
17 (71)	POLLEN	*Tetragonisca angustula* (Jataí)	SABOURAUD	*Schizosaccharomyces pombe* DSM 70577 DSM	NCBI:txid 4896	2842	(+++)
18 (23)	POLLEN	*Tetragonisca angustula* (Jataí)	SABOURAUD	*Kazachstania telluris* CBS 6265 CBS	NCBI:txid 36907	2892	(+++)
19 (11)	POLLEN	*Tetragonisca angustula* (Jataí)	SABOURAUD	*Kazachstania telluris* CBS 6265 CBS	NCBI:txid 36907	2842	(+++)
20 (103)	POLLEN	*Tetragonisca angustula* (Jataí)	YEAST EXTRACT	*Candida norvegica* DSM 70863 DSM	NCBI:txid 49330	2301	(+++)
21 (52)	POLLEN	*Tetragonisca angustula* (Jataí)	YEAST EXTRACT	*Candida maltosa* DSM 15531 DSM	NCBI:txid 5479	2091	(++)
22 (23)	POLLEN	*Tetragonisca angustula* (Jataí)	MALT EXTRACT	*Kazachstania telluris* CBS 6265 CBS	NCBI:txid 36907	2892	(+++)
23 (28)	HONEY	*Tetragonisca angustula* (Jataí)	SABOURAUD	*Schizosaccharomyces pombe* DSM 70577 DSM	NCBI:txid 4896	2892	(+++)
24 (42)	HONEY	*Tetragonisca angustula* (Jataí)	SABOURAUD	*Schizosaccharomyces pombe* DSM 70577 DSM	NCBI:txid 4896	2441	(+++)
25 (52)	HONEY	*Tetragonisca angustula* (Jataí)	SABOURAUD	*Scheffersomyces insectosus* CBS 4286 CBS	NCBI:txid 45590	2053	(++)
26 (127)	HONEY	*Tetragonisca angustula* (Jataí)	SABOURAUD	*Kazachstania telluris* CBS 2685T CBS	NCBI:txid 36907	2842	(+++)
27 (18)	HONEY	*Tetragonisca angustula* (Jataí)	YEAST EXTRACT	*Schizosaccharomyces pombe* DSM 70577 DSM	NCBI:txid 4896	2892	(+++)
28 (21)	HONEY	*Tetragonisca angustula* (Jataí)	MALT EXTRACT	*Kazachstania telluris* CBS 2685T CBS	NCBI:txid 36907	2342	(+++)
29 (227)	HONEY	*Melipona scutellaris* (Uruçu)	SABOURAUD	*Kazachstania telluris* CBS 2685T CBS	NCBI:txid 36907	2342	(+++)
30 (92)	HONEY	*Melipona scutellaris* (Uruçu)	YEAST EXTRACT	*Schizosaccharomyces pombe* DSM 70577 DSM	NCBI:txid 4896	2892	(+++)
31 (104)	HONEY	*Melipona scutellaris* (Uruçu)	MALT EXTRACT	*Brettanomyces bruxellensis* DSM 70742 DSM	NCBI:txid 5007	2653	(+++)
Total number of microbial isolates—2837 No colonies were found for *Melipona scutellaris* (Uruçu) pollen (+++)—highly probable species identification (++)—secure genus identification, probable species identification https://www.ncbi.nlm.nih.gov/taxonomy and https://www.mycobank.org/							

## Data Availability

Data will be made available on request.

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
