# Peer review of "Yeast Diversity in Honey and Pollen Samples from Stingless Bees in the State of Bahia, Brazil: Use of the MALDI-TOF MS/Genbank Proteomic Technique"

_microorganisms, 2024, doi:10.3390/microorganisms12040678_

Round 1
Reviewer 1 Report
Comments and Suggestions for Authors
Comments/Suggestions to the authors:
Manuscript Number: Microorganisms-2874855
Title: Yeast diversity in honey and pollen samples from stingless 2 bees in the state of Bahia, Brazil: Use of the MALDI-TOF 3 MS/Genbank proteomic technique
Dear authors:
This study focus on the study on the biodiversity of yeasts associated with honey and pollen of the stingless bee species from BA state, Brazil. The following comments/suggestions could be noticed by authors:
1. The scientific name of yeast and insect species should be in italic, as line 198, 202-208, 242-244, 301-305, 342-344. Please check that throughout the manuscript.
2. Some of the yeast species name in this manuscript should be updated according to the published reports and reliable taxonomy website, as Candida guilliermondii to Meyerozyma guilliermondii, Candida lactis-condensi to Starmerella lactis-condensi, and others else. Please check all the species name in this manuscript and revise them.
3. The yeast species names of yeast strains isolated from pollen and honey were identified based on MALDI-TOF as Fig. 2, Fig. 3, and Table 1, but the identification results were shown as strains (species name + strain number ). So only species names without strain number are suggested to represent the identification results.
4. The cell numbers of respective yeast species were counted to 8.5 Log CFU/g-1 (Fig. 3, text in line 221-22), and the count is as almost high as saturated cell concentration in pure culture of respective strains. Please check the accuracy of these data.
5. Some photos attached on the UPGMA dendrogram to represent the colony morphology of some species clustered in a same branch, but the colony morphology is not the typical one for all the strains in the clade. So the photos are suggested removed.
6. All the strain numbers or colony numbers are suggested to be labeled for each sample used on construction of the dendrogram, not only a species name to include all the strains.
7. All the yeast strains were identified only by MALDI-TOF in this manuscript, but the method is not standard for yeast identification. Molecular identification is suggested to be used as authentication of the identification for each species by MALDI-TOF.
8. In the section of materials and methods, all the detail condition and analysis parameters in the operation of MALDI-TOF are suggested to be listed.
9. Relationship or connection of microbiota between honey and pollen from different species of bee are suggested to explain based on the data of this study.
Author Response
Thank you very much for taking the time to review this manuscript. Please find the detailed responses below and the corresponding revisions/corrections highlighted/in track changes in the re-submitted files.
- The scientific name of yeast and insect species should be in italic, as line 198, 202-208, 242-244, 301-305, 342-344. Please check that throughout the manuscript. Thank you for your observation. This has been corrected throughout the manuscript.
- Some of the yeast species name in this manuscript should be updated according to the published reports and reliable taxonomy website, as Candida guilliermondii to Meyerozyma guilliermondii, Candida lactis-condensi to Starmerella lactis-condensi, and others else. Please check all the species name in this manuscript and revise them. We conducted the research for this taxonomic update and made this clear and referenced in the manuscript: “Some yeast species identified in this study received a taxonomic update according to https://www.mycobank.org/ - Candida lactis-condensi (Starmerella lactis-condensi); Candida shehatae var. insect (Scheffersomyces insectosus); Candida guilliermondii (Meyerozyma guilliermondii); Brettanomyces bruxelensis (or Dekkera bruxellensis)”.
- The yeast species names of yeast strains isolated from pollen and honey were identified based on MALDI-TOF as Fig. 2, Fig. 3, and Table 1, but the identification results were shown as strains (species name + strain number ). So only species names without strain number are suggested to represent the identification results. Table 1 shows "Morphotype code (number of isolates)" in its first column. A total of 31 morphotypes were coded. Among these codes, a similar numerical variety was also separated for identification, totaling 2,837. These are the codes of the isolates (Table 1).
- The cell numbers of respective yeast species were counted to 8.5 Log CFU/g-1 (Fig. 3, text in line 221-22), and the count is as almost high as saturated cell concentration in pure culture of respective strains. Please check the accuracy of these data. The data is correct. We first count each morphotype. After this count we purify the colony in successive "compound streaks". We later confirmed the morphotype and identified the isolate and returned to the initial count. This technique is standard in counting and isolating microorganisms.
- Some photos attached on the UPGMA dendrogram to represent the colony morphology of some species clustered in a same branch, but the colony morphology is not the typical one for all the strains in the clade. So the photos are suggested removed. We appreciate your observation. We corrected Figure 4 and deleted the images.
- All the strain numbers or colony numbers are suggested to be labeled for each sample used on construction of the dendrogram, not only a species name to include all the strains. We constructed the dendrograms using the spectra generated by MALDI-TOF. The software we use (UPGMA - BioNumerics software v. 7.10) performs statistical similarity grouping and projects the grouped result. Therefore, it is not possible to identify which spectrum of each isolate was used, because the program groups (up to thousands of spectra) in similarity for each microbial species.
- All the yeast strains were identified only by MALDI-TOF in this manuscript, but the method is not standard for yeast identification. Molecular identification is suggested to be used as authentication of the identification for each species by MALDI-TOF. We appreciate the reviewer's observation. We performed molecular extraction of total protein from each isolate (methodology in the manuscript). Subsequently, we used the isolated proteins for MALDI-TOF analysis and compared them with Genbank spectra. "NCBI Identifier" is referenced in Table 1. This molecular protein analysis methodology is currently used and accepted by the scientific community.
- In the section of materials and methods, all the detail condition and analysis parameters in the operation of MALDI-TOF are suggested to be listed. All methodology was detailed in the manuscript. We detail the beginning of protein isolation, the preparation phase for MALDI-TOF analysis; the analysis of the spectra; GenBank comparison and statistical construction of the dendrogram.
- Relationship or connection of microbiota between honey and pollen from different species of bee are suggested to explain based on the data of this study. The discussion topic of this manuscript has been improved.
Reviewer 2 Report
Comments and Suggestions for Authors
Dear Authors,
This paper identified the diversity of yeasts in honey and pollen by MALT-TOFMS.
Authors should revise the manuscript according to the comments.
1. Line157: Is an eppendorf tube a 1.5mL tube?
As you know, Eppendorf is the name of the company.
If it is an Eppendorf tube that Kasvi handles, it should be labeled as 1.5mL tube (Eppendorf).
2. 3.1. Yeast count:
How did the authors perform yeast species identification?
If the yeast species was identified by MALDI-TOFMS, the authors need to write the MALDI-TOFMS results first.
3. Through the manuscript:
Scientific names should be in italics.
4. Discussions:
The authors would do well to mention the safety of the isolated yeast.
Author Response
Thank you very much for taking the time to review this manuscript. Please find the detailed responses below and the corresponding revisions/corrections highlighted/in track changes in the re-submitted files.
This paper identified the diversity of yeasts in honey and pollen by MALT-TOFMS.
Authors should revise the manuscript according to the comments. Ok, corrected.
- Line157: Is an Eppendorf tube a 1.5mL tube? As you know, Eppendorf is the name of the company.
If it is an Eppendorf tube that Kasvi handles, it should be labeled as 1.5mL tube (Eppendorf). Corrected in text.
- 3.1. Yeast count:
How did the authors perform yeast species identification?
If the yeast species was identified by MALDI-TOFMS, the authors need to write the MALDI-TOFMS results first.
- Through the manuscript:
We first performed the plating and counting of yeast colonies. Subsequently, identification by MALDI-TOF was performed. For this reason, we first present the microbial count with the yeast species corresponding to each colony morphotype found. All the steps for identification by the MALDI-TOF technique are detailed in the manuscript.
Scientific names should be in italics. Corrected in text.
- Discussions:
The authors would do well to mention the safety of the isolated yeast. All yeast species identified in this manuscript were discussed in the text regarding origin and safety in relation to consumption.
Reviewer 3 Report
Comments and Suggestions for Authors
The submitted manuscript deals with the diversity and quantity of yeasts present in pollen and honey found in the hives of three stingless bee species. The authors isolated a huge number of yeast strains (more than 2800) and identified them using MALDI-TOF MS analysis. They found nine yeast species in the pollen samples and seven species in honey samples.
The manuscript contains interesting data, however, it requires significant improvement. The sentences throughout the article are too long and, therefore, difficult to read. I also strongly recommend to correct the English.
Page 2, lines 48 – 52: Pollen and nectar are important resources for the development, maintenance of the offspring and colony growth. Pollen is a being natural source of proteins, lipids, minerals, and vitamins, in addition to being as well as is a rich source of phytochemicals, which include flavonoids and phenolic acids [5,6]. Nectar is a main…
Page 2, lines 54 – 55: This pollen… The sentence is not clear, do you mean that bees produce some secretions rich in enzymes, or microorganisms produce enzymes, or both bees and microorganisms produce enzymes?
Page 2, line 66: Do you mean that honey produced by Apis bee does not contain bacteria and yeasts?
Page 2, line 70: nectar and pollen are products? Please correct this matter throughout the article.
Page 2, lines 73 – 75: In previous sentences you have mentioned stingless bee. In the sentence “Yeasts associated with…” you have mentioned stingless bee products (do you mean honey?) Yeasts associated with stingless bee products can be found in plants and flowers, as well as in pollen and nectar collected by them (pollen and nectar are present in flowers).
As well as in honey, fermented pollen (samburá), propolis, brood, in larvae, and in the digestive tract of bees (something is missing)
Page 2, lines 77 – 79. I recommend to move this sentence after the sentence: These microorganisms can act… (lines 89 – 91).
Page 2, lines 80 – 88: Rewrite these sentences, they are not clearly stated.
Page 3, lines 103 – 105: You have not studied the interaction between honey or pollen and yeasts. You have investigated the diversity of yeasts present in pollen and honey found in the hives of stingless bees.
Materials and methods
How many samples have you inspected?
Page 4, lines 127 - 135: A volume of 0.1 mL of samples were plated (100 μL or 0.1 mL) in onto three different culture media: Peptone Dextrose Yeast Extract 130 Agar (YEPD - agar 15 g/L, bacteriological peptone 20 g /L, dextrose 20 g/L, Yeast extract 10 g/L; Sigma, USA), Malt Extract Agar (YM - agar 15 g/L, bacteriological peptone 5 g/L, malt 30 g/L; Acumedia, USA) and Sabouraud Dextrose Agar (SDA - agar 15 g/L, bacteriological peptone 10 g/L, dextrose 40 g/L; Merck, Germany), all three media? supplemented with ampicillin (10 mg/L; Neo Química, Brazil) [33]. The samples were plated in triplicate.
Page 4, lines 136 – 138: Subsequently, the plates were incubated at 28 °C for five days. Colonies that showed typical similar characteristics were counted as representatives of individual species? The cell counts were expressed as log CFU.gˉ1.
Page 4, line 142: What is the depletion technique in microbiology?
Page 5, lines 172 – 177: An aliquot of 1 μL of the supernatant was applied to the 96-well MALDI-TOF MS target plate (Bruker Daltonics, Bremen, Germany), in triplicate, and allowed to dry in the environment at ambient (laboratory) (room) temperature. The sample was covered with 1 μL of the matrix (saturated solution of α-cyano-4-174 hydroxy-cinnamic acid (Sigma, USA) in 50% acetonitrile (Sigma, USA) and 2.5% trifluoroacetic acid (Merck, Germany), and after drying, the sample was measured using spectrometer (what type?, at which parameters? – e.g. mode).
3. Results
3.1. Yeast count Quantity and diversity of yeasts
You have examined diversity and abundance of yeasts present in honey and pollen samples, not the growth of yeasts. You have only interpreted the results obtained as the number of yeast colonies growing.
Page 5, line 210: Figure 2 shows that the highest number of yeasts was found in the pollen of Nannotrigona testaceicornes, with the predominant species…
Page 7, line 232 and thereafter: The samples of pollen and honey did not share yeast species. Similar species were found in both samples.
Page 7, lines 245 – 250. This part should be moved to “Discussion” section.
Page 7, lines 257 – 261 The description of the ranges of the logarithmic score in relation to the accuracy of the identification should be moved to the "Material and Methods" section (the ranges are given by the manufacturer Bruker Daltonic).
General comment: Please rewrite the entire "Results" section and avoid repeating results.
Rewrite the legend to Figures 2 and 3. Figure 2 (pollen is not from stingless bees).
Please check the current name of yeast species: Candida shehatae var. insectosa (current name is Scheffersomyces insectosus), Candida guilliermondii (current name is Meyerozyma guilliermondii), Brettanomyces bruxelensis (synonym Dekkera bruxellensis), Kazachstania exigua, Candida lactis-condensi (www.mycobank.com).
Figure 4. Taxonomically similar MALDI-TOF MS Spectra, grouped according to similarity by Pearson's correlation coefficient (UPGMA - BioNumerics software v. 7.10).
Discussion
Lines 278 – 283: Rosa et al… the species Starmerella neotropicalis…were also found in the pollen collected by stingless bees?
Lines 292 – 300 Please remove this paragraph it is unnecessary.
Lines 301 – 305: Check the current name of yeast species and put them in brackets
Change the position of the paragraph starting at line 301 with that starting at line 306.
Line 311: delete the word microbial
Line 342: The species listed by the authors are not new; they were found for the first time in the samples that the authors inspected.
The paragraph between lines 342 – 346 is redundant among the list of species and their properties. Information on the biotechnological potential of yeasts associated with pollen and honey of stingless bees should be moved to the end of the discussion or left only in the "Conclusions".
Please check references, some of them are not given correctly (e.g. the title of a reference is missing).
Please correct the "Abstract" based on the corrections you will be making throughout the manuscript.
Comments on the Quality of English Language
Author Response
Thank you very much for taking the time to review this manuscript. Please find the detailed responses below and the corresponding revisions/corrections highlighted/in track changes in the re-submitted files.
The submitted manuscript deals with the diversity and quantity of yeasts present in pollen and honey found in the hives of three stingless bee species. The authors isolated a huge number of yeast strains (more than 2800) and identified them using MALDI-TOF MS analysis. They found nine yeast species in the pollen samples and seven species in honey samples.
The manuscript contains interesting data, however, it requires significant improvement. The sentences throughout the article are too long and, therefore, difficult to read. I also strongly recommend to correct the English. English writing has been revised.
Page 2, lines 48 – 52: Pollen and nectar are important resources for the development, maintenance of the offspring and colony growth. Pollen is a being natural source of proteins, lipids, minerals, and vitamins, in addition to being as well as is a rich source of phytochemicals, which include flavonoids and phenolic acids [5,6]. Nectar is a main… This paragraph has been corrected in the manuscript.
Page 2, lines 54 – 55: This pollen… The sentence is not clear, do you mean that bees produce some secretions rich in enzymes, or microorganisms produce enzymes, or both bees and microorganisms produce enzymes? “microbial fermentation” This sentence has been corrected in the manuscript.
Page 2, line 66: Do you mean that honey produced by Apis bee does not contain bacteria and yeasts? The sentence was corrected in the manuscript to "a wider variety of microorganisms (bacteria and yeast)".
Page 2, line 70: nectar and pollen are products? Please correct this matter throughout the article. When referring to pollen, the word product was replaced by substance.
Page 2, lines 73 – 75: In previous sentences you have mentioned stingless bee. In the sentence “Yeasts associated with…” you have mentioned stingless bee products (do you mean honey?) Yeasts associated with stingless bee products can be found in plants and flowers, as well as in pollen and nectar collected by them (pollen and nectar are present in flowers). This has been corrected in the manuscript.
As well as in honey, fermented pollen (samburá), propolis, brood, in larvae, and in the digestive tract of bees (something is missing) Corrected “Yeasts associated/found with stingless bees can be found in plants and flowers (pollen and nectar) [20], as well as in the digestive tract of bees (secretions), in fermented pollen (samburá), in the honey and propolis produced and even in larvae” [21].
Page 2, lines 77 – 79. I recommend to move this sentence after the sentence: These microorganisms can act… (lines 89 – 91). The sentence has been removed from the manuscript.
Page 2, lines 80 – 88: Rewrite these sentences, they are not clearly stated. This paragraph has been corrected in the manuscript.
Page 3, lines 103 – 105: You have not studied the interaction between honey or pollen and yeasts. You have investigated the diversity of yeasts present in pollen and honey found in the hives of stingless bees. This has been corrected in the manuscript to “yeast diversity”.
Materials and methods
How many samples have you inspected? This has been added to the manuscript (10 samples of 20mL of honey and 15g of pollen from each beehive analyzed)
Page 4, lines 127 - 135: A volume of 0.1 mL of samples were plated (100 μL or 0.1 mL) in onto three different culture media: Peptone Dextrose Yeast Extract 130 Agar (YEPD - agar 15 g/L, bacteriological peptone 20 g /L, dextrose 20 g/L, Yeast extract 10 g/L; Sigma, USA), Malt Extract Agar (YM - agar 15 g/L, bacteriological peptone 5 g/L, malt 30 g/L; Acumedia, USA) and Sabouraud Dextrose Agar (SDA - agar 15 g/L, bacteriological peptone 10 g/L, dextrose 40 g/L; Merck, Germany), all three media? supplemented with ampicillin (10 mg/L; Neo Química, Brazil) [33]. The samples were plated in triplicate. Corrected to: All culture media were supplemented with ampicillin (10 mg/L; Neo Química, Brazil) [33]
Page 4, lines 136 – 138: Subsequently, the plates were incubated at 28 °C for five days. Colonies that showed typical similar characteristics were counted as representatives of individual species? The cell counts were expressed as log CFU.gˉ1. The count was performed by similarity of typical yeast colonies. Subsequently, the morphotypes were identified by the square root (√) of each morphotype.
Page 4, line 142: What is the depletion technique in microbiology? Added: Successive replication by compound streaks.
Page 5, lines 172 – 177: An aliquot of 1 μL of the supernatant was applied to the 96-well MALDI-TOF MS target plate (Bruker Daltonics, Bremen, Germany), in triplicate, and allowed to dry in the environment at ambient (laboratory) (room) temperature. The sample was covered with 1 μL of the matrix (saturated solution of α-cyano-4-174 hydroxy-cinnamic acid (Sigma, USA) in 50% acetonitrile (Sigma, USA) and 2.5% trifluoroacetic acid (Merck, Germany), and after drying, the sample was measured using spectrometer (what type?, at which parameters? – e.g. mode). This was not analyzed using a spectrophotometer. This was analyzed in the Maldi-TOF apparatus and spectra (sequence of peaks) were obtained. This was clarified in the text.
the plate was inserted into the equipment (MALDI-TOF MS) to obtain the spectra [38]. … and after drying, the plate was inserted into the equipment (MALDI-TOF MS) to obtain the spectra (sequence of peaks) [38].
- Results
3.1. Yeast count Quantity and diversity of yeasts
You have examined diversity and abundance of yeasts present in honey and pollen samples, not the growth of yeasts. You have only interpreted the results obtained as the number of yeast colonies growing. We counted yeast colonies present in the studied samples. This is described in the text.
Page 5, line 210: Figure 2 shows that the highest number of yeasts was found in the pollen of Nannotrigona testaceicornes, with the predominant species… The sentence was corrected in the manuscript.
Page 7, line 232 and thereafter: The samples of pollen and honey did not share yeast species. Similar species were found in both samples. Corrected.
Page 7, lines 245 – 250. This part should be moved to “Discussion” section. This has been done.
Page 7, lines 257 – 261 The description of the ranges of the logarithmic score in relation to the accuracy of the identification should be moved to the "Material and Methods" section (the ranges are given by the manufacturer Bruker Daltonic). Dear reviewer, we appreciate this observation, however we cannot move this data to methodology, because these data are results generated by the algorithm after analyzing the spectra.
General comment: Please rewrite the entire "Results" section and avoid repeating results. The results topic has been revised.
Rewrite the legend to Figures 2 and 3. Figure 2 (pollen is not from stingless bees). The caption for Figure 2 has been corrected to: "Figure 2. Yeast identified in the beehive pollen from studied stingless bees." Figure 3 is about the honey produced by bees.
Please check the current name of yeast species: Candida shehatae var. insectosa (current name is Scheffersomyces insectosus), Candida guilliermondii (current name is Meyerozyma guilliermondii), Brettanomyces bruxelensis (synonym Dekkera bruxellensis), Kazachstania exigua, Candida lactis-condensi (www.mycobank.com). We conducted the research for this taxonomic update and made this clear and referenced in the manuscript: “Some yeast species identified in this study received a taxonomic update according to https://www.mycobank.org/ - Candida lactis-condensi (Starmerella lactis-condensi); Candida shehatae var. insect (Scheffersomyces insectosus); Candida guilliermondii (Meyerozyma guilliermondii); Brettanomyces bruxelensis (or Dekkera bruxellensis)”.
Figure 4. Taxonomically similar MALDI-TOF MS Spectra, grouped according to similarity by Pearson's correlation coefficient (UPGMA - BioNumerics software v. 7.10). Caption of the figure has been simplified to: Figure 4. Taxonomically similar MALDI-TOF MS Spectra.
Discussion
Lines 278 – 283: Rosa et al… the species Starmerella neotropicalis…were also found in the pollen collected by stingless bees? Yes, this has been corrected in the manuscript
Lines 292 – 300 Please remove this paragraph it is unnecessary. The discussion topic has been modified.
Lines 301 – 305: Check the current name of yeast species and put them in brackets We conducted the research for this taxonomic update and made this clear and referenced in the manuscript: “Some yeast species identified in this study received a taxonomic update according to https://www.mycobank.org/ - Candida lactis-condensi (Starmerella lactis-condensi); Candida shehatae var. insect (Scheffersomyces insectosus); Candida guilliermondii (Meyerozyma guilliermondii); Brettanomyces bruxelensis (or Dekkera bruxellensis)”.
Change the position of the paragraph starting at line 301 with that starting at line 306. The discussion topic has been modified.
Line 311: delete the word microbial The discussion topic has been modified.
Line 342: The species listed by the authors are not new; they were found for the first time in the samples that the authors inspected. This has been corrected in the manuscript.
The paragraph between lines 342 – 346 is redundant among the list of species and their properties. Information on the biotechnological potential of yeasts associated with pollen and honey of stingless bees should be moved to the end of the discussion or left only in the "Conclusions". This has been corrected in the manuscript.
Please check references, some of them are not given correctly (e.g. the title of a reference is missing). Ok, revised.
Please correct the "Abstract" based on the corrections you will be making throughout the manuscript. Ok, corrected.